# Ultrasound image segmentation based on Transformer and U-Net with joint loss

Lina Cai[1], Qingkai Li[2], Junhua Zhang[3], Zhenghua Zhang[4], Rui Yang[5] and Lun Zhang[1]

[1] Yunnan Vocational Institute of Energy Technology, Qujing, Yunnan, China
[2] Department of Radiology, First Hospital of Qujing, Qujing, Yunnan, China
[3] School of Information Science and Engineering, Yunnan University, Kunming, Yunnan, China
[4] Medical Imaging Department, First Affiliated Hospital of Kunming Medical University, Kunming, Yunnan, China
[5] Qujing Power Supply Bureau, Qujing, Yunnan, China

## ABSTRACT

**Background:** Ultrasound image segmentation is challenging due to the low signal-to-noise ratio and poor quality of ultrasound images. With deep learning advancements, convolutional neural networks (CNNs) have been widely used for ultrasound image segmentation. However, due to the intrinsic locality of convolutional operations and the varying shapes of segmentation objects, segmentation methods based on CNNs still face challenges with accuracy and generalization. In addition, Transformer is a network architecture with self-attention mechanisms that performs well in the field of computer vision. Based on the characteristics of Transformer and CNNs, we propose a hybrid architecture based on Transformer and U-Net with joint loss for ultrasound image segmentation, referred to as TU-Net.

**Methods:** TU-Net is based on the encoder-decoder architecture and includes encoder, parallel attention mechanism and decoder modules. The encoder module is responsible for reducing dimensions and capturing different levels of feature information from ultrasound images; the parallel attention mechanism is responsible for capturing global and multiscale local feature information; and the decoder module is responsible for gradually recovering dimensions and delineating the boundaries of the segmentation target. Additionally, we adopt joint loss to optimize learning and improve segmentation accuracy. We use experiments on datasets of two types of ultrasound images to verify the proposed architecture. We use the Dice scores, precision, recall, Hausdorff distance (HD) and average symmetric surface distance (ASD) as evaluation metrics for segmentation performance.

**Results:** For the brachia plexus and fetal head ultrasound image datasets, TU-Net achieves mean Dice scores of 79.59% and 97.94%; precisions of 81.25% and 98.18%; recalls of 80.19% and 97.72%; HDs (mm) of 12.44 and 6.93; and ASDs (mm) of 4.29 and 2.97, respectively. Compared with those of the other six segmentation algorithms, the mean values of TU-Net increased by approximately 3.41%, 2.62%, 3.74%, 36.40% and 31.96% for the Dice score, precision, recall, HD and ASD, respectively.



Corresponding author
Lun Zhang, zhanglun1104@163.com

## INTRODUCTION

As a medical imaging modality, ultrasound imaging has been widely applied in clinical screening, diagnosis and treatment. Accurately segmenting ultrasound images is very important for making subsequent diagnoses. Unlike computer tomography (CT) and magnetic resonance imaging (MRI), ultrasound imaging is portable, cost-effective and uses nonionizing radiation. Nevertheless, due to their coherent nature, ultrasound images are impacted by speckle noise, missing boundaries and low signal-to-noise ratios (SNR). Therefore, ultrasound images are more difficult to segment than other medical images (*Fiorentino et al., 2023*; *Wang et al., 2021*). Some algorithms used in traditional image segmentation have been applied to ultrasound images, but they have not improved segmentation accuracy. With deep learning advancements, some approaches based on convolutional neural networks (CNNs) have been widely used in the field of ultrasound imaging. In particular, architectures based on the encoder-decoder architecture, such as U-Net, have improved segmentation accuracy (*Malhotra et al., 2022*; *Ronneberger, Fscher & Brox, 2015*). The encoder module is responsible for reducing dimensions and capturing different levels of feature information. The decoder module is responsible for gradually recovering dimensions and delineating boundaries of the segmentation target. In addition, the skip connection between the encoder and decoder can compensate for the loss of feature information caused by successive convolutions and pooling. However, due to the intrinsic locality of convolution operations, these approaches are limited by global context. The attention mechanism can be used as a resource allocation scheme, which is the main method for addressing information overload and is applied in the computer vision field (*de Santana Correia & Colombini, 2022*). Hence, some researchers have combined attention mechanisms with CNNs to capture global feature information.

Transformers can capture global feature information with their long-range dependency capabilities. Therefore, the Transformer-based model has achieved state-of-the-art performance in natural language processing (NLP). The subsequently proposed Vision Transformer (ViT) applied image recognition to improve results. ViT takes image patches as input and uses self-attention to learn the global information of all image patches. Some approaches based on ViT applied image segmentation and improved performance. However, ViT focuses on global feature information and lacks localization information. Similarly, some approaches based on encoder-decoder apply successive convolutions and pooling, resulting in a lack of global feature information and spatial information. Due to the various shapes, sizes and blurry boundaries of ultrasound images, segmenting objects is difficult. Considering these issues, we propose a hybrid Transformer and U-Net with joint loss (TU-Net) for the segmentation of ultrasound images. The contributions of this work can be summarized as follows: (1) TU-Net can integrate local and global feature information with CNNs and Transformers. (2) We adopt the parallel attention mechanism in the proposed TU-Net. Among the attention mechanisms, the one based on CNNs can extract multiscale local feature information of targets with varying shapes, and the one based on Transformer can capture global feature information. (3) For segmentation objects with blurry boundaries, we propose using the Dice and TopK joint loss to improve

prediction accuracy. We validate the effectiveness of our proposed method; the proposed method outperforms state-of-the-art methods on ultrasound images of the brachial plexus (BP) and fetal head.

The remainder of the article is organized as follows. The related works are shown in "Related Work". The proposed method and implementation details are explored in "Materials and Methods". Extensive experiments are conducted to evaluate our proposed methods in "Experimental Details". In "Results" and "Discussion", we discuss these results and conclude this article.

# RELATED WORK

## Segmentation networks based on CNNs

With the advancement of deep learning, some approaches based on CNNs are widely used in the field of ultrasound imaging segmentation. Based on U-Net, *Zhou et al. (2019)* proposed using a series of dense skip pathways to capture more feature information from images. This method can compensate for the loss of feature information caused by successive convolutions and pooling. Due to the intrinsic locality of convolution operations, approaches based on U-Net are limited by global context. Therefore, some studies have adopted multiple-channel convolution to solve this problem. *Mehta & Sivaswamy (2017)* proposed a novel network that used CNNs to combine and represent 3D context information for brain structure segmentation (M-Net). *Javaid, Dasnoy & Lee (2018)* proposed dilated convolution with U-Net to extract global feature information for breast image segmentation (dilated U-Net). *Zhang et al. (2020)* proposed a multiple-channel with a large kernel convolution network for ultrasound image segmentation (MA-Net). When segmenting objects with various shapes and sizes, these approaches based on CNNs have weak generalization.

## Segmentation networks based on Transformer

The transformer model originally designed for NLP and the subsequently proposed ViT applied image recognition to achieve better results. Meanwhile, some approaches based on ViT have been applied in the field of image segmentation. *Wang et al. (2022)* proposed a novel mixed transformer module for simultaneous intra- and inter-affinity learning for medical image segmentation. *Shen et al. (2022)* applied Transformer with residual axial attention for breast structure segmentation of ultrasound images. *Gao, Zhou & Metaxas (2021)* proposed an efficient self-attention mechanism along with relative position encoding for medical image segmentation. *Chen et al. (2021)* supported both Transformer and U-Net for medical image segmentation. *Zhang, Liu & Hu (2021)* combined Transformer and CNNs in parallel to capture global dependency and low-level feature information. Inspired by these approaches, we propose a hybrid Transformer and U-Net architecture with a parallel attention mechanism. It can integrate CNNs and Transformer to better perform ultrasound image segmentation.

## MATERIALS AND METHODS

The proposed TU-Net includes encoder, parallel attention mechanism and decoder modules. The structure of TU-Net is shown in Fig. 1. First, the ultrasound image is input into a successive encoder module to obtain high-dimensional feature maps. Next, these feature maps are input into the parallel attention mechanism to obtain global and multiscale local feature information. Finally, the different level feature information in the encoder module is connected to the decoder module by a skip pathway to generate the segmentation mask.

### Encoder and decoder modules

Networks based on the encoder-decoder architecture have been widely used in image segmentation. The encoder module is responsible for reducing dimensions and capturing different levels of feature information of ultrasound images. The decoder module is responsible for gradually recovering dimensions and delineating the boundaries of the segmentation target. In the proposed TU-Net, ResNet-50 is used as the encoder to capture the feature information of the input image. The decoder module consists of an upsampling layer and two successive convolution layers, which are used to recover spatial dimensions and boundary information. Due to the loss of feature information caused by successive convolutions and pooling, we adopt skip pathways to deliver feature information captured by the encoder to the decoder (*Drozdzal et al., 2016*).

### Parallel attention mechanism module

Many networks based on the attention mechanism have been widely applied in the field of image segmentation. The attention mechanism can avoid using multiple similar feature maps and focus on the most salient and informative features without additional supervision. Recently, ViT has achieved excellent performance in many computer vision tasks using a self-attention mechanism. However, it focuses on global feature information and neglects localization information. Similarly, some approaches based on encoder-decoder apply successive convolutions and pooling, resulting in a lack of global feature information and a loss of spatial information. Therefore, we adopt the self-attention mechanism of Transformer to capture the global feature information and use a series of atrous convolution and pyramid pooling to capture multiscale local feature information.

In traditional segmentation networks based on ViT, Transformer is directly used as an encoder to extract feature information from images. As shown by some experiments and studies by other researchers, this method cannot improve segmentation accuracy (*Chen et al., 2021*). Therefore, we utilize the high-dimensional feature maps from the successive encoder module as the input of the transformer module. We first reshape the input feature maps into a sequence of flattened two-dimensional (2D) patches. Each patch is of size $P \times P$ ($P = 16$), and the number of image patches is $N = H \times W/P^2$ ($H$ and $W$ specify the dimensions of feature maps). Second, we use a trainable linear projection to map the patches into a latent D-dimensional embedding. To retain the positional information of

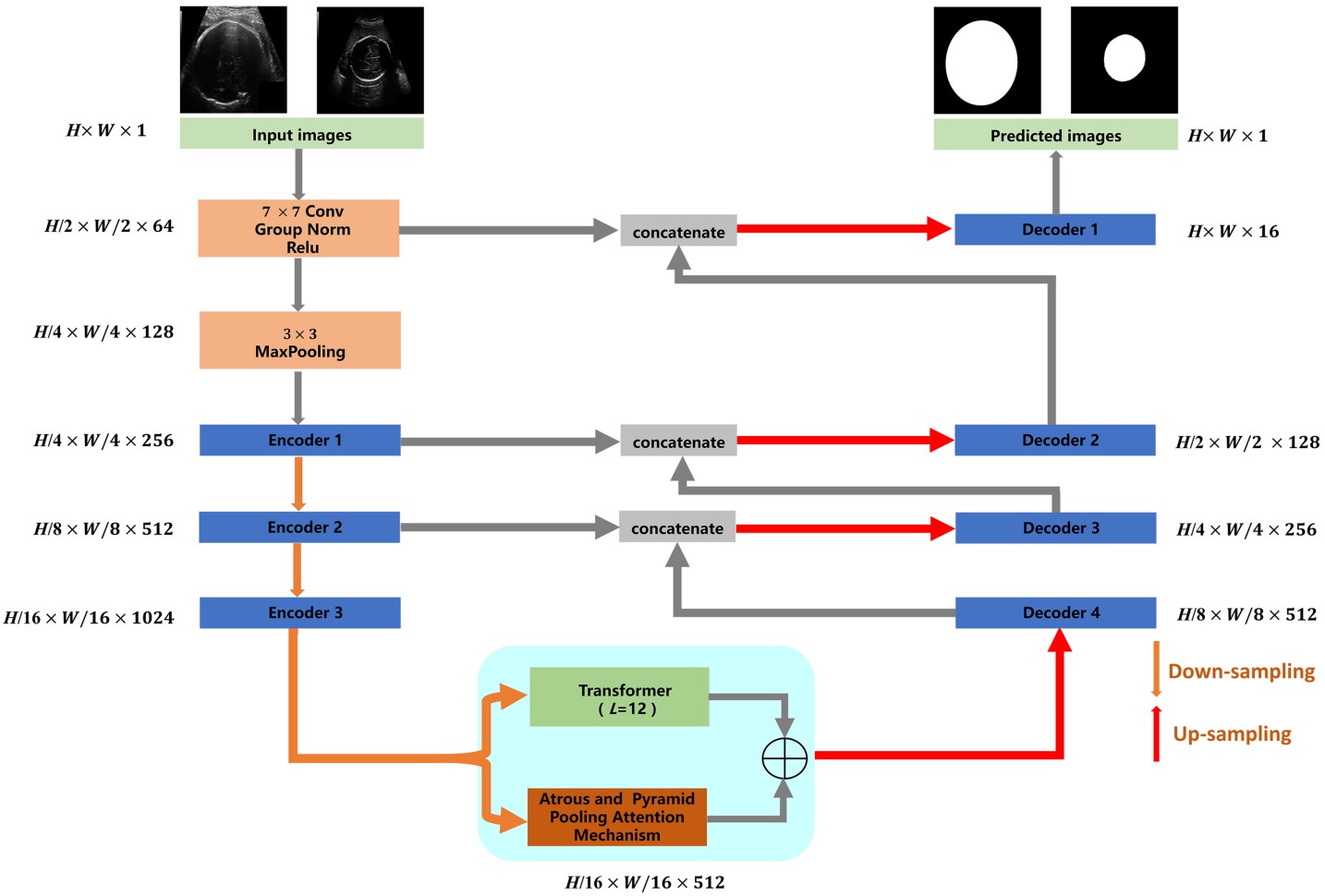

**Figure 1  Architecture of TU-Net.** $H \times W \times C$ represents the dimensions of each module ($C$ = 1, 64, 128, 256, 512, 1,024). Image source credit: *van den Heuvel et al. (2018a)*. Automated measurement of fetal head circumference using 2D ultrasound images (Data set). Zenodo. https://doi.org/10.5281/zenodo.1327317, https://hc18.grand-challenge.org.

the patches, we use position embeddings that are added to the embeddings of the patches. The self-attention mechanism of TU-Net consists of $L$ ($L$ = 12) transformer modules. Each transformer module consists of a multihead self-attention (MSA) layer and a multilayer perceptron (MLP) layer (*Dosovitskiy et al., 2020*). These layers are connected in turn. The output of each transformer module can be written as follows:

$$z_l' = MSA(LN(z_{l-1})) + z_{l-1} \qquad l = 1 \ldots . L \qquad (1)$$
$$z_l = MLP(LN(z_l')) + z_l' \qquad l = 1 \ldots . L \qquad (2)$$

where $LN(.)$ denotes the layer normalization operator, and $Z_l$ is the transformer image representation. The structure of the transformer module is illustrated in Fig. 2.

To address boundary information loss and segmentation objects with various sizes, we adopt a series of atrous convolution and pyramid pooling modules to capture multiscale
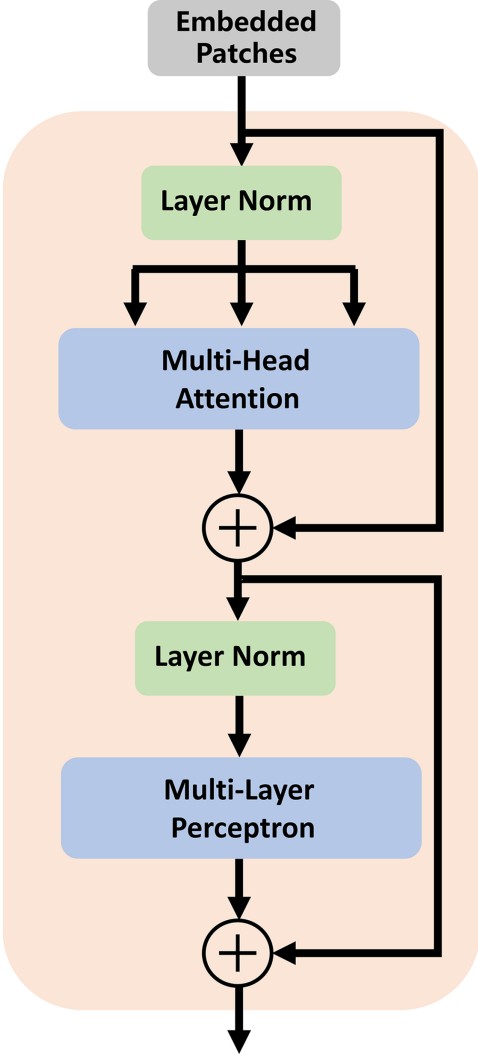

**Figure 2** **Architecture of the transformer module.** The self-attention mechanism consists of $L$ ($L = 12$) transformer modules.

local feature information. The architecture of the atrous convolution module is shown in Fig. 3A. It has four cascade atrous convolution branches. The four atrous convolutions have different sampling strides ($r = 1, 3, 5$ and $7$). In the end, four atrous convolution branches are combined as the input of the pyramid pooling module. Subsequently, we adopt the pyramid pooling module to detect objects with various sizes. The architecture of the pyramid pooling module is shown in Fig. 3B. First, the module has four cascade pooling layers with four receiving fields of $2 \times 2$, $4 \times 4$, $8 \times 8$ and $16 \times 16$. Second, the output of each pooling layer goes through a $1 \times 1$ convolution to reduce the dimensions of the feature maps. Subsequently, the upsampling layer is used to restore the feature map to its original size. Finally, four outputs of the upsampling layers are combined as the input of the decoder module.

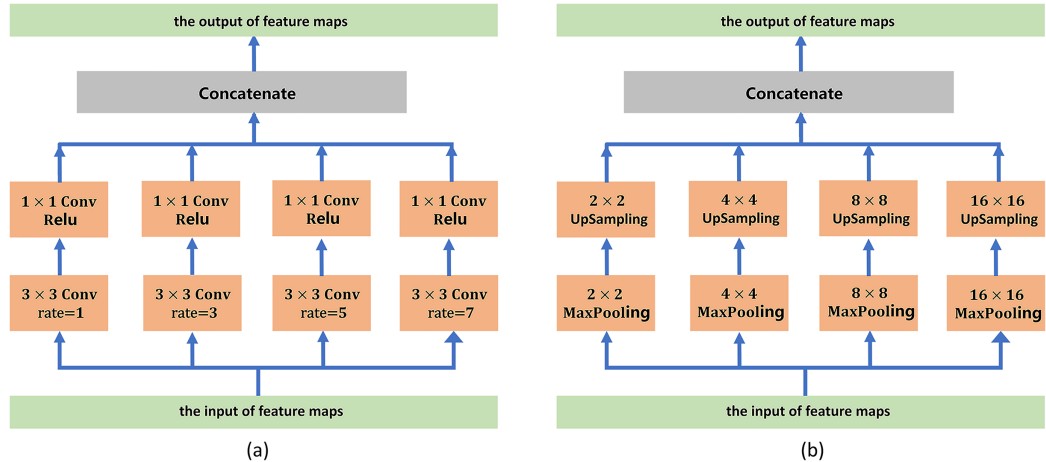

**Figure 3** (A) Architecture of the atrous convolution module. (B) Architecture of the pyramid pooling module.

## Joint loss

Image segmentation is used to determine whether a pixel belongs to the foreground or background of an image. The Dice coefficient is the most commonly used segmentation evaluation metric in the field of image segmentation. This coefficient represents the overlapping region of the ground truth and the prediction. The Dice coefficient is given as:

$$Dice = \frac{2|X \cap Y|}{|X| + |Y|} \tag{3}$$

where $|X|$ indicates the ground-truth pixels, $|Y|$ represents the value predicted pixels and $|X \cap Y|$ is the value of shared pixels in $|X|$ and $|Y|$.

The Dice loss function is defined as follows:

$$Dice_{loss} = 1 - Dice \tag{4}$$

However, the size of the regions of interest in different ultrasound images greatly vary. Thus, the learning process can become trapped in local minima of the loss function, resulting in the predictions of the network being strongly biased towards the background and missing or only partially detecting the foreground. Meanwhile, most labelled pixels can easily be discriminated against others and further research on these pixels will not improve the accuracy. Therefore, we add TopK loss in the process of training. TopK loss is also a variant of cross entropy, which is used to force networks towards hard samples and automatically balance biased training data during processing (*Ma et al., 2021*).

The TopK loss function is defined as:

$$L_{TopK} = -\frac{1}{N} \sum_{c=i}^{c} \sum_{i \in K} g_i^c \log s_i^c \tag{5}$$

where $g_i$ and $s_i$ denote the ground truth and predicted segmentation of voxel $i$, respectively. $C$ is the number of classes, and $N$ is the number of voxels. $K$ is the set of the $k\%$ worst pixels. In TU-Net, the TopK loss with $k = 10\%$ is the default setting.

In the proposed TU-Net, we adopt joint loss to optimize learning and improve prediction accuracy. $L_{total}$ is the sum of the Dice and TopK loss functions. The joint loss function is given as:

$$L_{total} = L_{Dice} + L_{Topk} \tag{6}$$

where $L_{Dice}$ and $L_{Topk}$ are the Dice and TopK loss functions, respectively.

## Evaluation metrics

We also use precision, recall, Hausdorff distance (HD) and average symmetric surface distance (ASD) as evaluation metrics in addition to the Dice scores. Precision and recall are defined according to Eqs. (7) and (8), respectively.

$$precision = \frac{TP}{TP + FP} \tag{7}$$

$$recall = \frac{TP}{TP + FN} \tag{8}$$

where TN, TP, FN and FP are the true-negative, true-positive, false-negative and false-positive values, respectively (*Chang et al., 2009*). Because there are considerable noise and outliers in the ultrasound images, we also include metrics based on the surface distance. The HD represents the maximum value of misalignment between two objects, which is mainly used to evaluate the structural difference between two targets. Smaller HD values represent higher segmentation accuracy (*Zhang et al., 2020*).

The sets of points of $A$ and $B$ are $S(A)$ and $S(B)$, respectively. The shortest distance of $S(A)$ to an arbitrary point $v$ is given by:

$$d(S(A), v) = \min_{S_A \in S(A)} \|S_A - v\| \tag{9}$$

The shortest distance of $S(B)$ to an arbitrary point $v$ is given by:

$$d(S(B), v) = \min_{S_B \in S(B)} \|S_B - v\| \tag{10}$$

where $\|\cdot\|$ represents the Euclidean distance. The HD is defined as:

$$HD(A, B) = max\left( \max_{v \in S(A)} \min_{S_B \in S(B)} \|S_B - v\|, \max_{v \in S(B)} \min_{S_A \in S(A)} \|S_A - v\| \right) \tag{11}$$

The ASD represents the average value of surface distances of A to B and B to A. Similarly, smaller ASD values represent higher segmentation accuracy. The ASD is defined as:

$$ASD(A, B) = \frac{1}{|S(A)| + |S(B)|} \left( \sum_{S_A \in S(A)} d(S_A, S(B)) + \sum_{S_B \in S(B)} d(S_B, S(A)) \right) \tag{12}$$

**Table 1  The hyperparameters of algorithms.**

| Hyper parameter | U-Net | U-Net++ | M-Net | Dilated U-Net | MA-Net | TransUNet | TU-Net |
|---|---|---|---|---|---|---|---|
| Loss functions | Dice | Dice & CE | Dice & CE | CE | Dice & CE | Dice & CE | Dice & TopK |
| Batch size | 8 | 4 | 4 | 10 | 4 | 24 | 10 |
| Optimizer | SGD | Adam | Adam | Adam | SGD | SGD | SGD |
| Learning rate | 0.001 | 3E–4 | 0.00001 | 0.001 | 0.001 | 0.01 | 0.01 |
| Momentum | 0.95 | None | None | None | 0.9 | 0.9 | 0.9 |

**Note:**
CE and Dice represent the cross-entropy and dice loss functions, respectively.

## EXPERIMENTAL DETAILS

PyTorch 1.7.0 was applied as the framework to train TU-Net. The optimizer adopts mini-batch stochastic gradient descent (SGD) with a weight decay of 0.0001 and momentum of 0.9. The initial learning rate was set to 0.01 and fine-tuned every 100 epochs. The workstation used to train TU-Net was a 2080Ti graphic card with 11 GB of memory. The proposed TU-Net is compared with other image segmentation algorithms, including U-Net (*Ronneberger, Fscher & Brox, 2015*), U-Net++ (*Zhou et al., 2019*), M-Net (*Mehta & Sivaswamy, 2017*), dilated U-Net (*Javaid, Dasnoy & Lee, 2018*), MA-Net (*Zhang et al., 2020*) and TransUNet (*Chen et al., 2021*). These algorithms are derived from their applications in the original works, and the hyperparameters used for training are given in Table 1. In addition, the Friedman test is applied as a statistical analysis to evaluate the segmentation performance of the algorithms. In this analysis, 5% is a significant level.

## RESULTS

To demonstrate the superiority of TU-Net segmentation performance, we compare it to other segmentation algorithms, including U-Net, U-Net++, M-Net, dilated U-Net, MA-Net and TransUNet. We use the mean value and standard deviation of Dice scores, precision, recall, HD and ASD to evaluate algorithm performance. Finally, SPSS 23.0 is used for statistical analysis of the above algorithms.

### Branchia Plexus datasets

Because BP is an important motor and sensory nerve of the upper limb, blocking BP can relieve much pain in upper limb surgery. Therefore, accurately segmenting the structure of BP is very important for anaesthesia during upper limb surgery. The BP datasets of ultrasound images are taken from the 2016 Kaggle competition (*Montoya et al., 2016*; *Zhang & Zhang, 2022*). This dataset includes segmentation objects with various sizes. Because the test datasets were not released in the competition, the collected training datasets are randomly divided into training datasets and test datasets in our experiments. The training dataset includes 1,710 samples, and the test dataset includes 448 samples. Because the number of training samples is small, we apply horizontal flipping, vertical flipping, random scaling and rotation to increase the number of training samples. Ultimately, the number of training samples is 8,550. Meanwhile, all samples are cropped to $320 \times 320$ for our experiments.

**Table 2 The mean and standard deviation of five evaluation metrics for BP datasets.**

| Method | Dice (%) | Precision (%) | Recall (%) | HD (mm) | ASD (mm) |
|---|---|---|---|---|---|
| U-Net | 74.53 ± 0.19 | 74.73 ± 0.19 | 77.66 ± 0.22 | 16.70 ± 14.85 | 5.78 ± 5.92 |
| U-Net++ | 74.76 ± 0.19 | 75.48 ± 0.19 | 77.03 ± 0.22 | 16.53 ± 14.36 | 5.69 ± 5.77 |
| M-Net | 75.94 ± 0.19 | 78.11 ± 0.19 | 76.16 ± 0.21 | 14.48 ± 14.03 | 5.45 ± 7.49 |
| dilated U-Net | 75.89 ± 0.19 | 79.16 ± 0.19 | 75.18 ± 0.21 | 14.57 ± 13.21 | 5.27 ± 5.54 |
| MA-Net | 77.55 ± 0.17 | 78.65 ± 0.18 | 78.82 ± 0.19 | 13.59 ± 13.03 | 4.82 ± 4.57 |
| TransUNet | 79.15 ± 0.16 | 80.40 ± 0.17 | 79.93 ± 0.18 | 12.89 ± 11.96 | 4.51 ± 2.87 |
| TU-Net | **79.59 ± 0.16** | **81.25 ± 0.17** | **80.19 ± 0.18** | **12.44 ± 11.23** | **4.29 ± 2.57** |

**Note:**
The best result is highlighted with bold.

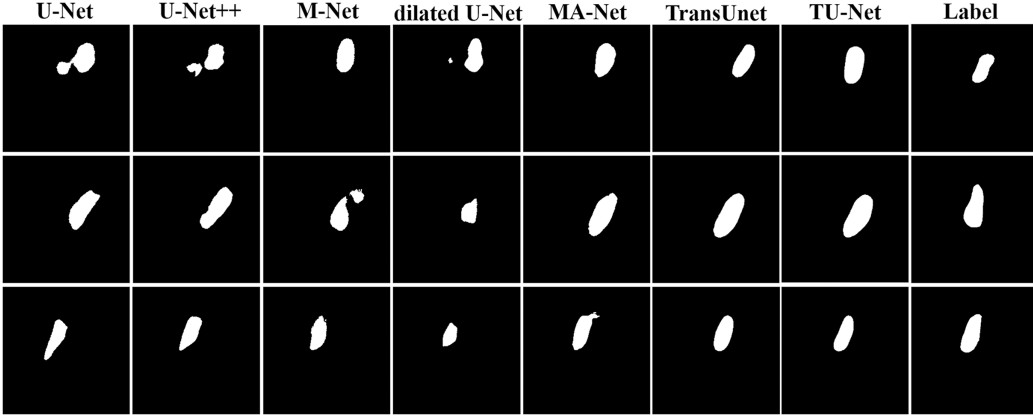

**Figure 4 Segmentation results of different algorithms on BP datasets.**

Table 2 shows the mean and standard deviation of five evaluation metrics for different segmentation algorithms. TU-Net achieves a value of 79.59 ± 0.16 for Dice (%), 81.25 ± 0.17 for precision (%), 80.19 ± 0.18 for recall (%), 12.44 ± 11.23 for HD (mm) and 4.29 ± 2.57 for ASD (mm). The segmentation accuracy of our proposed method is better than that of the other methods. The segmentation results of the seven networks are shown in Fig. 4.

## Fetal head datasets

Genetic factors and nutrition intake can affect the growth of fetus. Obstetricians generally monitor fetal health and development by measuring the head circumference of foetuses from ultrasound images. Therefore, the accuracy of head structure segmentation affects the accuracy of head circumference measurement. In the subsequent experiments, the fetal head image datasets are from the medical image segmentation challenge (*van den Heuvel et al., 2018b*). It contains 788 ultrasound images of fetal heads at various stages. These images are randomly divided into training and test datasets. The training datasets include 718 samples, and the test datasets include 70 samples. Because the number of training samples is small, we apply horizontal flipping, vertical flipping, random scaling and rotation to increase the number of training samples. Ultimately, the number of training samples is 3,590. Meanwhile, all samples are cropped to 320 × 320 for our experiments.

**Table 3 The mean and standard deviation of five evaluation metrics for the fetal head datasets.**

| Method | Dice (%) | Precision (%) | Recall (%) | HD (mm) | ASD (mm) |
|---|---|---|---|---|---|
| U-Net | 92.67 ± 0.07 | 98.03 ± 0.01 | 88.66 ± 0.11 | 56.62 ± 41.26 | 12.50 ± 7.93 |
| U-Net++ | 93.59 ± 0.05 | 98.12 ± 0.01 | 90.00 ± 0.09 | 50.85 ± 40.95 | 11.26 ± 7.83 |
| M-Net | 95.68 ± 0.03 | 96.96 ± 0.02 | 94.70 ± 0.05 | 19.26 ± 21.96 | 5.51 ± 3.56 |
| Dilated U-Net | 96.76 ± 0.02 | 97.09 ± 0.02 | 96.54 ± 0.04 | 16.48 ± 19.24 | 5.24 ± 3.59 |
| MA-Net | 97.26 ± 0.01 | 96.79 ± 0.02 | **97.78 ± 0.02** | 10.92 ± 5.49 | 4.15 ± 1.62 |
| TransUNet | 97.73 ± 0.01 | 98.15 ± 0.01 | 97.35 ± 0.02 | 7.99 ± 2.84 | 3.29 ± 1.19 |
| TU-Net | **97.94 ± 0.01** | **98.18 ± 0.01** | 97.72 ± 0.01 | **6.93 ± 2.15** | **2.97 ± 0.94** |

Note:
The best result is highlighted with bold.

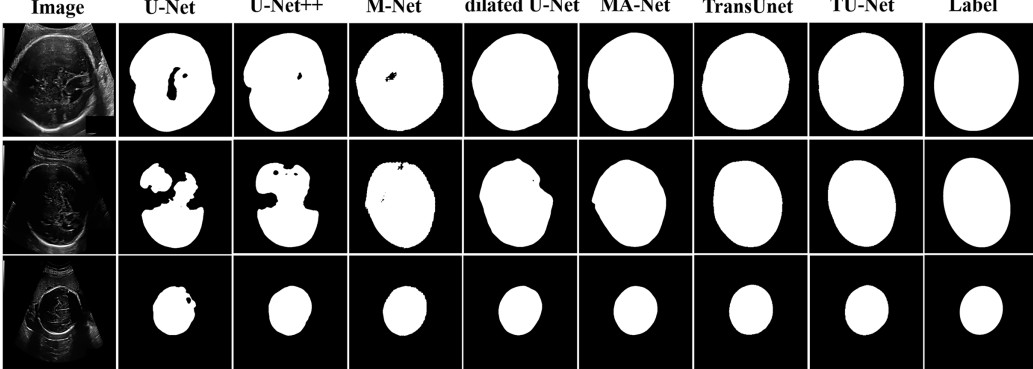

**Figure 5 Segmentation results of different algorithms on fetal head datasets.** Image source credit: *van den Heuvel et al. (2018a)*. Automated measurement of fetal head circumference using 2D ultrasound images (Data set). Zenodo. https://doi.org/10.5281/zenodo.1327317, https://hc18.grand-challenge.org.

Table 3 shows the mean and standard deviation of five evaluation metrics for different segmentation algorithms. TU-Net achieves a value of 97.94 ± 0.01 for Dice (%), 98.18 ± 0.01 for precision (%), 97.72 ± 0.01 for recall (%), 6.93 ± 2.15 for HD (mm) and 2.97 ± 0.94 for ASD (mm). Our proposed method significantly improves segmentation accuracy. The segmentation results of the seven networks are shown in Fig. 5.

## Ablation study

To evaluate the effectiveness of each module in TU-Net, we conducted ablation studies on the two types of datasets. Dice scores are used as the evaluation metric of the following experiments.

## Joint loss

Using different loss functions with the same architecture of the network greatly impacts segmentation performance. Therefore, we adopt joint loss to optimize learning efficiency and improve segmentation accuracy. We test different combinations of loss functions to determine the optimal joint loss in the experiments (*e.g.*, Dice, Dice & CE, Dice & Focal, Dice & CE & TopK and Dice & TopK). Figure 6 displays boxplots for the mean Dice scores of different loss functions in TU-Net. These plots indicate that the median of our proposed
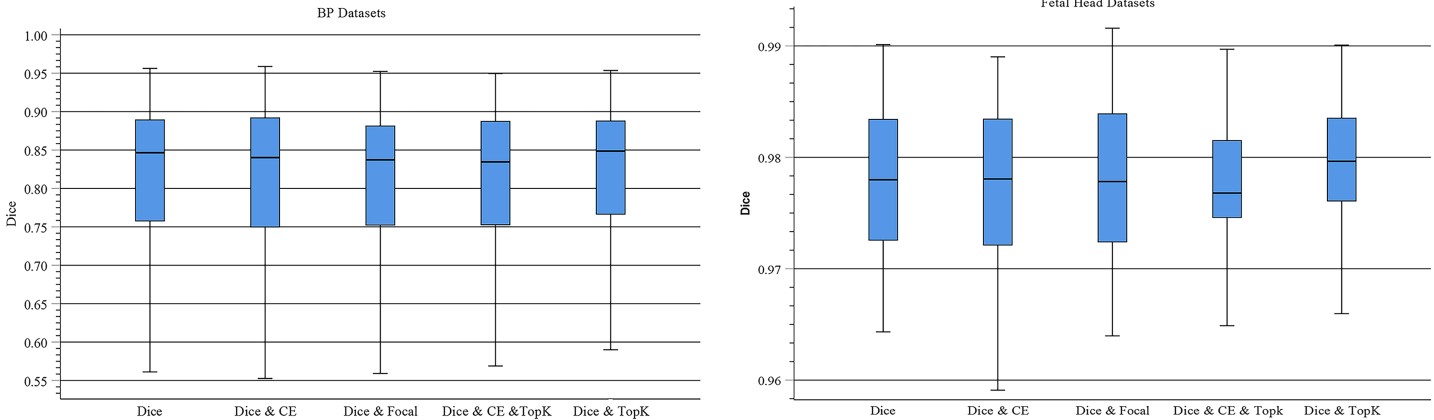

**Figure 6** **Boxplots of dice scores for different joint losses on BP and fetal head datasets.** The central mark indicates the median, the bottom and top edges of the box indicate the interquartile range and the whiskers indicate the minimum and maximum.

methods is the maximum median. Meanwhile, we observe that segmentation performance is strongly dependent on the combination loss function and that the Dice-related compound loss function has better segmentation accuracy than the other functions. Therefore, the compound loss function of Dice and TopK is the optimal loss function. Compared with other loss functions, our proposed method can improve the mean Dice values of the BP and fetal head datasets by approximately 1.08% and 0.17%, respectively.

## Parallel attention mechanism module

TU-Net adopts a parallel attention mechanism to capture the global and multiscale local feature information of ultrasound images. The parallel attention mechanism consists of a transformer module and a series of atrous convolution and pyramid pooling modules. To demonstrate the advantages of the parallel attention mechanism in TU-Net, we conduct the following experiments on the BP and fetal head datasets. First, we remove the parallel attention mechanism from TU-Net (U-Net). Similarly, we remove the transformer module from the parallel attention mechanism in TU-Net (AU-Net). Finally, we remove the atrous convolution and pyramid module from the parallel attention mechanism in TU-Net (T-Net). Figure 7 displays boxplots for the mean Dice scores of different attention mechanism modules. Figure 7 indicates that the median of our proposed methods is the maximum and our proposed method can improve segmentation accuracy. These experiments indicate that the parallel attention mechanism module can improve the mean Dice scores of the BP and fetal head datasets by approximately 0.80% and 0.13%, respectively.

## Input feature

Traditionally, ViT takes image patches as the input of Transformer to learn the global relation of all image patches. In TU-Net, we first obtain high-dimensional feature maps from the successive encoder module. Next, these feature maps are input into the parallel attention mechanism to obtain global and multiscale local feature information. Finally, the

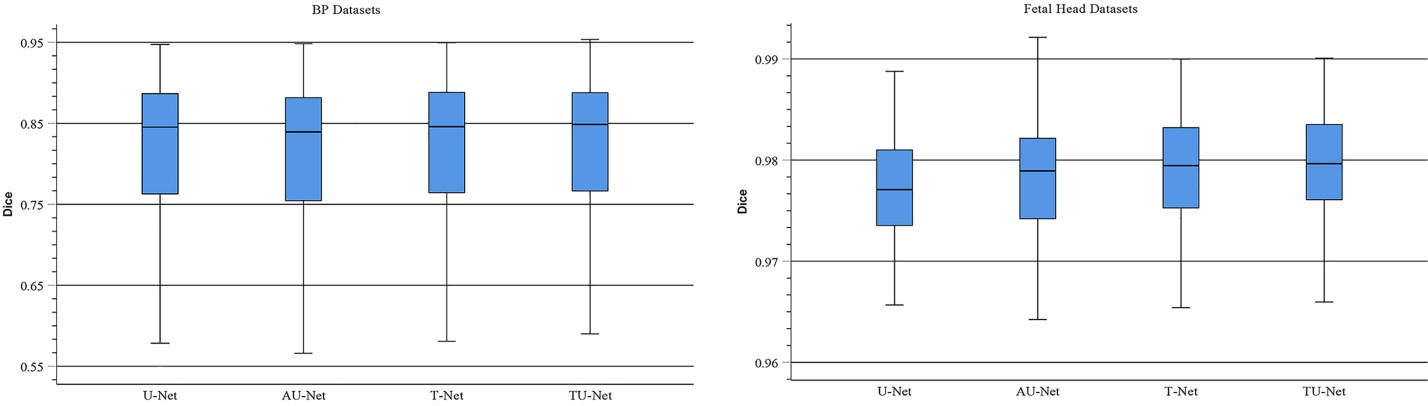

**Figure 7 Boxplots of dice scores for different attention mechanisms on BP and fetal head datasets.** The central mark indicates the median, the bottom and top edges of the box indicate the interquartile range and the whiskers indicate the minimum and maximum.

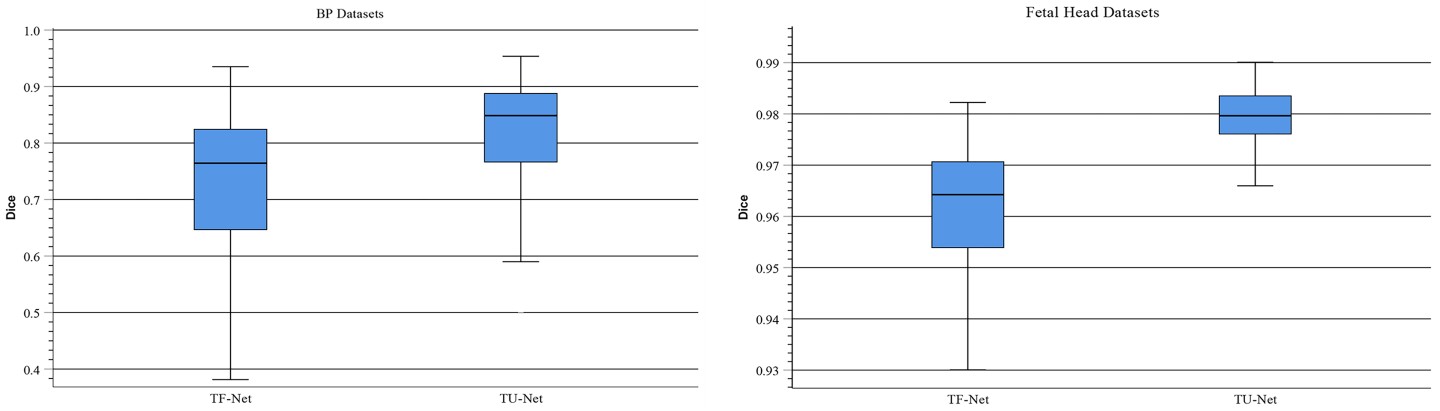

**Figure 8 Boxplots of dice scores for different input types on BP and fetal head datasets.** The central mark indicates the median, the bottom and top edges of the box indicate the interquartile range and the whiskers indicate the minimum and maximum.

skip connection between the encoder and decoder compensates for the loss of feature information caused by successive convolutions and pooling. Therefore, to demonstrate the advantages of high-dimensional feature maps, we conduct the following experiment. We remove the encoder and decoder modules and use image patches instead of high-dimensional feature maps as the input of the Transformer module (T-Net). Figure 8 displays boxplots for the mean Dice scores of the above experiment. This indicates that the median of our proposed methods is the maximum and that using high-dimensional feature maps as the input of Transformer can improve segmentation accuracy. Meanwhile, we observe that segmentation performance of different input types largely differ on different types of ultrasound images. Compared with T-Net, our proposed network can improve the mean Dice scores of the BP and fetal head datasets by approximately 13.01% and 1.96%, respectively.

**Table 4 The mean rank of the dice scores of different networks on BP and fetal head datasets.**

| Datasets | Mean rank | | | | | | | p-value |
|---|---|---|---|---|---|---|---|---|
| | U-Net | U-Net++ | M-Net | Dilated U-Net | MA-Net | TransUNet | TU-Net | |
| BP | 3.43 | 3.62 | 4.15 | 4.06 | 4.20 | 4.25 | **4.30** | 1.96E−13 |
| Fetal head | 1.86 | 2.03 | 3.21 | 4.40 | 4.69 | 5.60 | **6.21** | 5.27E−45 |

Note:
The best result is highlighted with bold.

## Statistical analysis

We apply statistical analysis to evaluate the performance of different networks. Since the Dice scores are not a Gaussian distribution, we use the nonparametric Friedman test to evaluate segmentation performance (*Friedman, 2012*). The mean rank and $p$-value are shown in Table 4. A $p$-value less than 0.05 was considered to indicate a significant difference across the compared algorithms. Higher mean rank values indicate higher segmentation performance. The results of the statistical analysis are shown in Table 4. These results show that TU-Net significantly improves segmentation performance compared with other algorithms.

## DISCUSSION

In this article, we propose a hybrid Transformer and U-Net with a joint loss algorithm for the segmentation of ultrasound images. The proposed algorithm is based on the encoder-decoder architecture and includes encoder, parallel attention mechanism and decoder modules. Meanwhile, we adopt a compound loss function with Dice and TopK to optimize learning efficiency and improve segmentation accuracy. Finally, we use comparison experiments and ablation studies to verify our proposed algorithm on two types of ultrasound image datasets.

In the comparison experiments, we compare two types of segmentation algorithms: one is segmentation algorithms based on CNNs, and the other is segmentation algorithms based on hybrid CNNs and Transformer. Figure 9 shows the bar plots of the mean metric scores of the seven algorithms. It can be observed that the hybrid CNNs and Transformer algorithms perform better than other algorithms based on CNNs because CNNs and Transformers can capture local and global feature information, respectively. This fusion of local and global feature information can improve segmentation accuracy. Meanwhile, in the ablation studies, we find the key components for improved segmentation performance. First, the high-dimensional feature maps obtained by the encoder module can be used as the input of the parallel attention mechanism to improve segmentation accuracy. Second, the parallel attention mechanism can capture the global and multiscale local feature information of ultrasound images. This method can address segmentation targets with various sizes, improving the generalization of the algorithm. Third, the Dice-related compound loss function has better segmentation accuracy than the other loss functions. However, our proposed method outperforms other methods on the segmentation tasks of ultrasound images, and the number of trainable parameters significantly increases. The bar plots of various algorithm parameter sizes are shown in Fig. 10. Therefore, we require more
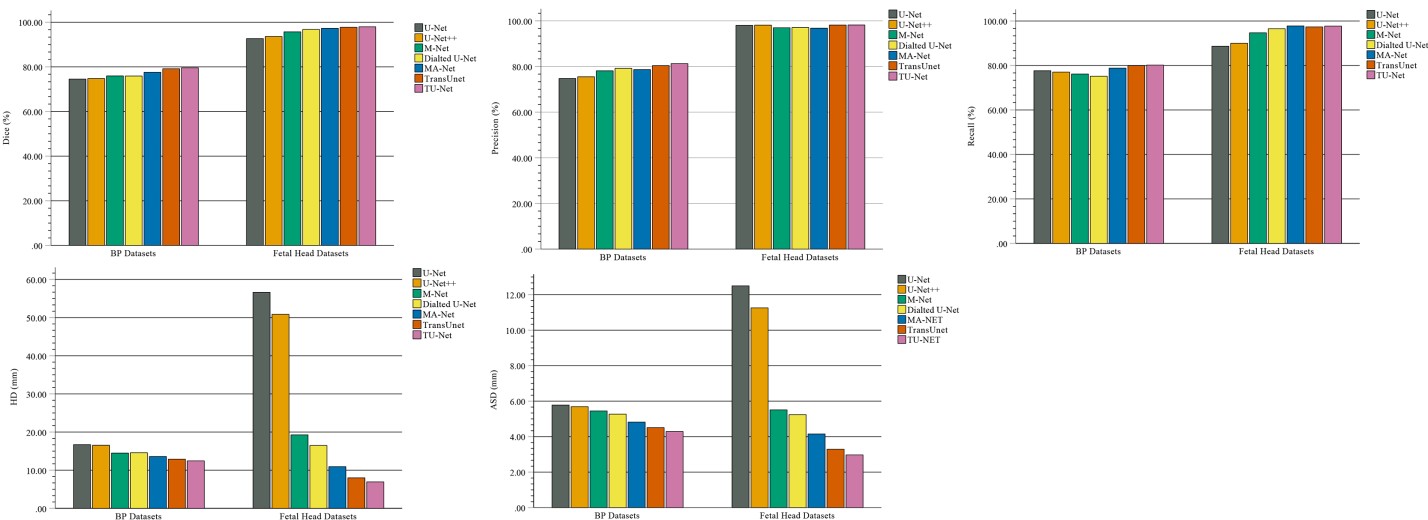

**Figure 9 Bar plots of evaluation metrics for seven segmentation algorithms.** The coloured bar represents the mean Dice scores, precision, recall, HD and ASD of each algorithm on the BP and fetal head datasets.

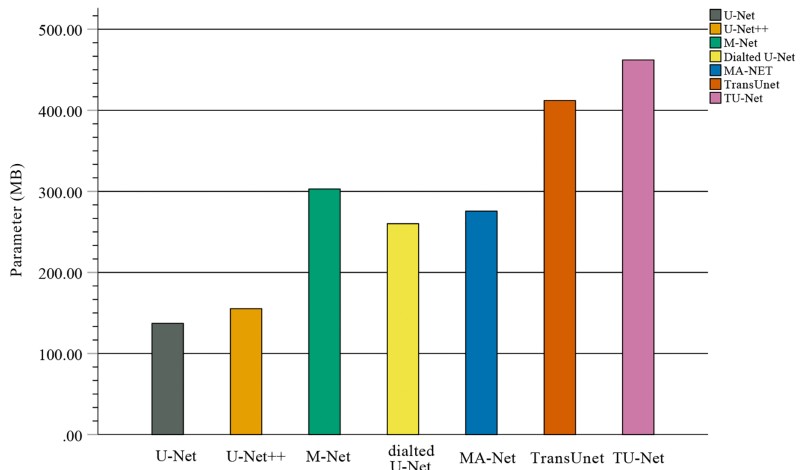

**Figure 10 Bar plots of parameter sizes for seven segmentation algorithms.**

computational resources for training the algorithms. However, we believe this problem can be alleviated as modern GPUs rapidly develop.

## CONCLUSIONS

In this article, we propose a hybrid Transformer and U-Net with joint loss to segment ultrasound images. TU-Net is based on the encoder-decoder architecture and includes three modules: the encoder, parallel attention mechanism and decoder modules. We use two types of ultrasound image datasets to verify our proposed method. As shown by comparative experiments, TU-Net significantly improves segmentation performance.

Compared with the other six algorithms, on average, TU-Net shows improvements of approximately 3.89%, 1.03%, 4.31%, 39.97% and 32.06% for Dice scores, precision, recall, HD and ASD, respectively. In addition, we verify the performance of different algorithms using the Freidman test of nonparametric statistical analysis. TU-Net obtains the best mean rank in this test. This result indicates that the different algorithms significantly differ. Meanwhile, we conducted a series of ablation studies to verify the effectiveness of the TU-Net. These experimental results show that the parallel attention mechanism, joint loss and input of feature maps can effectively improve segmentation accuracy.

## ACKNOWLEDGEMENTS

We would like to thank Bo Li and Yang Zhao for their constructive discussion during the manuscript revision. Meanwhile, we would like to thank Kaggle and Zenodo for providing the public datasets for the experiment.

### Funding

The research work was supported by the grants from the Natural Science Foundation of China under Grants 62063034. The funders had no role in study design, data collection and analysis, decision to publish, or preparation of the manuscript.

### Grant Disclosures

The following grant information was disclosed by the authors:
Natural Science Foundation of China: 62063034.

### Competing Interests

Rui Yang is an employee of the Qujing Power Supply Bureau. The authors declare that they have no competing interests.

### Author Contributions

- Lina Cai conceived and designed the experiments, performed the experiments, analyzed the data, performed the computation work, prepared figures and/or tables, authored or reviewed drafts of the article, and approved the final draft.
- Qingkai Li performed the experiments, analyzed the data, prepared figures and/or tables, and approved the final draft.
- Junhua Zhang conceived and designed the experiments, analyzed the data, authored or reviewed drafts of the article, and approved the final draft.
- Zhenghua Zhang performed the experiments, analyzed the data, prepared figures and/or tables, and approved the final draft.
- Rui Yang performed the experiments, analyzed the data, prepared figures and/or tables, and approved the final draft.
- Lun Zhang conceived and designed the experiments, analyzed the data, performed the computation work, prepared figures and/or tables, authored or reviewed drafts of the article, and approved the final draft.

## Data Availability

The code is available in the Supplemental File.

The datasets of the fetal head are available at Zenodo:

- Thomas L. A. van den Heuvel, Dagmar de Bruijn, Chris L. de Korte and Bram van Ginneken. Automated measurement of fetal head circumference using 2D ultrasound images. PLOS ONE 13.8 (2018): e0200412.

- Thomas L. A. van den Heuvel, Dagmar de Bruijn, Chris L. de Korte, & Bram van Ginneken. (2018). Automated measurement of fetal head circumference using 2D ultrasound images [Data set]. Zenodo. https://doi.org/10.5281/zenodo.1327317.

- https://hc18.grand-challenge.org/.

The datasets of the brachial plexus are available at Kaggle:

- Montoya A, Hasnin, Kaggle446, Shirzad, Cukierski W, Yffud. 2016. Ultrasound Nerve Segmentation. Kaggle.

- https://kaggle.com/competitions/ultrasound-nerve-segmentation.

## Supplemental Information

Supplemental information for this article can be found online at http://dx.doi.org/10.7717/peerj-cs.1638#supplemental-information.

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
