# Peer review of "Ultrasound image segmentation based on Transformer and U-Net with joint loss"

_PeerJ Computer Science, doi:10.7717/peerj-cs.1638_

## Round 0.1 · original submission · Major Revisions

Dear authors,

Your article has not been recommended for publication in its current form. However, we do encourage you to address the concerns and criticisms of the reviewers and resubmit your article once you have updated it accordingly.

Reviewer 1 ·

Basic reporting

Researchers have performed segmentation of ultrasound images in their study titled "Ultrasound image segmentation based on Transformer and U-Net with joint loss". Based on the characteristics of transformer and U-Net, a hybrid architecture based on common loss Transformer and U-Net is proposed for ultrasound image segmentation.

Experimental design

First of all, if we examine the article with the Abstract section, I would like to point out that the abstract is written too long. For example, ". In particular, the U-shape architecture (U-Net) has shown improved segmentation performance." There is no need to mention the shape of the U-net architecture in the sentence. For topic integrity, there should be a smooth transition between paragraphs in the Introduction section. The researchers performed data multiplexing in their study. No information is given about the original image numbers and the data numbers after multiplexing. In addition, it is necessary to provide information about the classes in the data set. While testing the models, were the images obtained by multiplexing used? Figure 5,6,7 should be detailed.

Validity of the findings

The proposed method should be compared with similar studies in the literature in the Introduction section. In addition, these studies should be presented in the Discussion section as a table. Confusion matrices can be used in the Result section. Limitations of the study should be presented. The Discussion section was like a summary of the Result section, this section should be rewritten.

Additional comments

Spelling errors and repetitive expressions should be reviewed.

Cite this review as

Reviewer 2 ·

Basic reporting

This paper proposes a hybrid Transformer and U-Net model named TU-Net for ultrasound image segmentation. TU-Net is based on an encoder-decoder architecture and uses Transformer as the attention mechanism to capture the global feature information. In general, the contribution of this work is limited and the experimental results are not convincing.

1. In my opinion, the Transformer encoder block approach has been used in many segmentation studies. The architecture proposed in this work is not substantially different from those that have appeared in previous work. Therefore, the technical novelty of this work seems to be limited.

2. In the abstract, the authors argue that "Ultrasound image segmentation is a challenging task due to its low signal-to-noise ratio and poor quality". How does the proposed approach address this challenge?

Experimental design

3. What is the difference between TU-Net and TransUnet[3]? Since the hybrid method of CNN-Transformer is utilized, it is not enough to only compare CNN methods in the comparison experiment, and it is necessary to compare the TransUnet [3] method.

4. In the ablation experiment, the authors' comparison experiment is not sufficient, and the results of the experiment with or without the Transformer should be set separately. Also, why use "atrous convolution to replace Transformer" instead of using spatial attention mechanism GA [1] or channel attention mechanism CA[2]? I think atrous convolution can not replace Transformer in ablation experiments.

Validity of the findings

no comment

Additional comments

5. The manuscript should add: "2. Related work".


[1] Oktay O , Schlemper J , Folgoc L L , et al. Attention U-Net: Learning Where to Look for the Pancreas[J]. 2018.
[2] J. Hu, L. Shen and G. Sun, "Squeeze-and-Excitation Networks," 2018 IEEE/CVF Conference on Computer Vision and Pattern Recognition, Salt Lake City, UT, USA, 2018, pp. 7132-7141, doi: 10.1109/CVPR.2018.00745.
[3] Chen J , Lu Y , Yu Q , et al. TransUNet: Transformers Make Strong Encoders for Medical Image Segmentation[J]. 2021.

Annotated reviews are not available for download in order to protect the identity of reviewers who chose to remain anonymous.
Cite this review as

---

## Round 0.2 · accepted · Accept

Dear authors,

Thank you for the revision. The paper seems to be improved in the opinion of the reviewers. The paper is now ready to be published.

Best wishes,

Reviewer 1 ·

Basic reporting

I thank the authors for their careful consideration of the shortcomings noted in the previous revision.

Experimental design

I thank the authors for their careful consideration of the shortcomings noted in the previous revision.

Validity of the findings

I thank the authors for their careful consideration of the shortcomings noted in the previous revision.

Additional comments

I thank the authors for their careful consideration of the shortcomings noted in the previous revision.

Cite this review as